# Robust Hybrid Quantum-Classical Latent Diffusion Models via Quantum Noise

## Abstract

While quantum generative models offer computational advantages, quantum noise, unavoidable in real quantum hardware, is typically viewed as an obstacle to performance. We challenge this perspective by demonstrating that controlled quantum noise can be an implicit regularizer for latent diffusion models. We introduce a hybrid quantum-classical latent diffusion architecture where quantum error channels are injected between parameterized quantum circuit (PQC) layers in the latent space during the forward pass. We maintain a Gaussian backward process to enable training efficiency on current quantum hardware. Theoretically, we prove these quantum channels control the model's Lipschitz constant with depth and shrink the quantum Fisher information matrix (QFIM), yielding flatter minima within the loss landscape and hence tighter PAC-Bayesian generalization bounds. We conduct extensive experiments on MNIST and CIFAR-10, comparing our model with existing baselines. Our noise-regularized models improve the Frechet inception distance (FID) by 18.7% over noiseless baselines while maintaining superior robustness under adversarial attacks and parameter perturbations. To our knowledge, this work is among the first to systematically study transforming unavoidable quantum noise into a leverage for robust generative models.

## 1 Introduction

The arrival of deep generative models has marked a transformative period in machine learning (ML), with architectures such as Denoising Diffusion Probabilistic Models (DDPMs) Ho et al. (2020) and Latent Diffusion Models (LDMs) Rombach et al. (2022) achieving state-of-the-art performance in data synthesis across various domains, including audio and image generation (Ho et al., 2020; Rombach et al., 2022). These models operate by systematically corrupting data with noise through a fixed forward process and then training a neural network to reverse this process, thus learning the underlying data distribution efficiently.

Concurrently, quantum computing has emerged as a promising paradigm with the potential to solve problems intractable for classical computers. The development of quantum machine learning (QML) seeks to leverage quantum computing (Biamonte et al., 2017) to enhance machine learning algorithms. However, the current Noisy Intermediate-Scale Quantum (NISQ) era is defined by the primary challenge of quantum decoherence (De Falco et al., 2025), which is the loss of quantum information due to the uncontrolled interactions with the environment. Due to decoherence, there have been several quantum error correction and mitigation techniques to combat the effects of this inherent noise (Preskill, 2018).

Taking these into account, we present a new hybrid Quantum Latent Diffusion Model (QLDM). The architecture integrates a classical Variational Autoencoder (VAE) (Kingma & Welling, 2013) with a quantum forward diffusion process. The VAE first encodes high-dimensional image data into a low-dimensional latent space. Within this space, a PQC, interleaved with tunable quantum error channels, executes the forward diffusion, mapping the latent representations to a noise distribution. Then, a classical score-based denoising network learns to execute the reverse process, thus generating novel data samples. This hybrid design maintains training efficiency on current hardware in the NISQ era by confining the quantum operations to the forward pass while leveraging the power of classical deep learning for the reverse generation.

We present several contributions in this work. Firstly, we propose a novel hybrid QLDM architecture that systematically leverages quantum channel noise as a core component of the generative process. Secondly, we establish a theoretical framework, proving that injecting quantum noise induces a contraction of the model's Lipschitz constant and quantum Fisher information matrix (QFIM). These results provide formal guarantees for enhanced input robustness, parameter robustness, and generalization. Thirdly, we provide comprehensive experimental validation on MNIST and CIFAR-10 datasets, demonstrating that the proposed noise-regularized QLDM achieves superior sample quality, adversarial and parameter robustness, and generalization capabilities when compared to both classical and noiseless quantum baselines.

## 2 RELATED WORK

**Classical Diffusion and Latent Diffusion** Diffusion probabilistic models generate data by learning to invert a Markov chain that gradually adds Gaussian noise to clean samples, a technique first explored by Sohl-Dickstein et al. (2015) and later refined by Ho et al. (2020). A neural network is trained to reverse this process and recover clean data. Although pixel–space diffusion achieves impressive sample quality, the approach is computationally expensive because every denoising step operates on high–dimensional image tensors. LDMs reduce this burden by training an autoencoder that maps images to a low-dimensional latent space where the diffusion process occurs, after which a decoder reconstructs the final image (Rombach et al., 2022). Operating in the latent domain preserves perceptual fidelity while reducing memory consumption and accelerating both training and inference. We adopt this latent-space strategy, adding quantum operations into the forward diffusion stage.

Denoising Diffusion Probabilistic Models (DDPMs) are a class of generative models that learn to produce data by reversing a gradual noising process (Sohl-Dickstein et al., 2015; Ho et al., 2020). The model consists of two Markov chains: a forward process and a reverse process. The forward process, $q$, is fixed and progressively adds Gaussian noise to a data sample $x_0$ over $T$ timesteps according to a variance schedule $\{\beta_t\}_{t=1}^T$: $q(x_t \mid x_{t-1}) = \mathcal{N}(x_t; \sqrt{1 - \beta_t}\, x_{t-1},\, \beta_t I)$.

A useful property of this process is that we can sample $x_t$ at any timestep $t$ directly from $x_0$: $q(x_t \mid x_0) = \mathcal{N}(x_t; \sqrt{\bar{\alpha}_t}\, x_0,\, (1 - \bar{\alpha}_t)I)$, where $\alpha_t = 1 - \beta_t$ and $\bar{\alpha}_t = \prod_{s=1}^t \alpha_s$. As $t \to T$, the distribution $q(x_T \mid x_0)$ approaches a standard isotropic Gaussian distribution, $\mathcal{N}(0, I)$.

The reverse process, $p_\theta$, is a learned Markov chain that aims to reverse the diffusion process, starting from pure noise $x_T \sim \mathcal{N}(0, I)$ and generating a data sample $x_0$. Each step of this process is parameterized by a neural network $\epsilon_\theta$: $p_\theta(x_{t-1} \mid x_t) = \mathcal{N}(x_{t-1}; \mu_\theta(x_t, t),\, \Sigma_\theta(x_t, t))$.

Training is performed by optimizing the evidence lower bound (ELBO) on the log-likelihood. A simplified training objective, which has been shown to be effective, is to train a network $\epsilon_\theta(x_t, t)$ to predict the noise $\epsilon$ that was added to create $x_t$: $\mathcal{L}_{\text{simple}}(\theta) = \mathbb{E}_{t, x_0, \epsilon}\left[\|\epsilon - \epsilon_\theta(\sqrt{\bar{\alpha}_t}\, x_0 + \sqrt{1 - \bar{\alpha}_t}\, \epsilon,\, t)\|^2\right]$, where $t$ is sampled uniformly from $\{1, \ldots, T\}$, $x_0 \sim q(x_0)$, and $\epsilon \sim \mathcal{N}(0, I)$.

A key limitation of DDPMs is the computational cost of operating in the high-dimensional pixel space. LDMs address this by applying the diffusion process in a compressed, lower-dimensional latent space learned by a VAE (Kingma & Welling, 2013). An encoder first maps an image $x$ to a latent representation $z$, the diffusion process operates on $z$, and a decoder then maps the denoised latent back to the pixel space. This approach significantly reduces computational requirements while achieving high-resolution image synthesis. The architecture proposed in this work adopts this latent-space strategy.

**Quantum Generative Models** In quantum mechanics, the state of a system is represented by a density matrix $\rho$, a positive semi-definite operator ($\rho \geq 0$) on a Hilbert space $\mathcal{H}$ with unit trace, $\text{Tr}(\rho) = 1$ (Nielsen & Chuang, 2010). Pure states correspond to projectors $\rho = |\psi\rangle\langle\psi|$, while mixed states describe statistical ensembles, necessary for modeling open systems interacting with an environment. The most general physical evolution is a quantum channel, a linear map $\mathcal{N}$ that is Completely Positive and Trace-Preserving (CPTP) (Preskill, 2018). Trace preservation ensures $\text{Tr}(\mathcal{N}(\rho)) = \text{Tr}(\rho)$, while complete positivity requires that $\mathcal{I}_A \otimes \mathcal{N}$ remains positive for any ancillary Hilbert space $\mathcal{H}_A$, guaranteeing physicality even when the system is entangled. Any CPTP map admits an operator-sum, or *Kraus*, representation, where its action on a state $\rho$ is given by

$\mathcal{N}(\rho) = \sum_k A_k \rho A_k^\dagger$. The $\otimes$ denotes the tensor product, and the operators $\{A_k\}$ are called Kraus operators and satisfy the trace-preserving condition $\sum_k A_k^\dagger A_k = I$, ensuring that $\mathrm{Tr}(\mathcal{N}(\rho)) = \mathrm{Tr}(\rho)$ (Nielsen & Chuang, 2010).

The intersection of quantum computing and generative modeling has produced several distinct approaches, including Quantum Generative Adversarial Networks (QGANs) (Lloyd & Weedbrook, 2018) and Quantum Circuit Born Machines (QCBMs) (Liu & Wang, 2018), which use quantum processors as generators/discriminators, and can provably converge to learning the data distribution (Lloyd & Weedbrook, 2018). QCBMs prepare parameterized quantum states whose measurement probabilities model a distribution, trained via kernels (Liu & Wang, 2018). However, these approaches are typically limited to small toy datasets or quantum-state data, and do not address modern diffusion methods.

More recently, hybrid quantum-classical diffusion models have been explored. Parigi et al. (2024) introduced Quantum-Noise-Driven Diffusion models, explicitly using quantum channels in the forward process. Their work emphasizes the interplay of coherence, entanglement, and noise, proposing hybrid classical-quantum (Classical-Quantum, Quantum-Classical, Quantum-Quantum) diffusion. Yet this was mostly a proof-of-concept without formal analysis of robustness or generalization. De Falco et al. (2025) propose Quantum Latent Diffusion Models with a classical autoencoder followed by variational circuits in latent space. They compare generated images quantitatively to classical diffusion, finding that their small-quantum model outperforms a similarly sized classical model, especially in few-shot settings. (Yeter-Aydeniz et al., 2025) study hybrid QLDMs for medical images, using qubit-residual blocks in the latent U-Net. They report higher-quality images (86% gradable vs 69% for classical) and find that noisy quantum models can even surpass classical ones when run on noisy hardware.

While our work frames quantum noise as a beneficial regularizer, this perspective must be reconciled with the substantial body of research identifying noise as a primary cause of the noise-induced barren plateau (NIBP) phenomenon (McClean et al., 2018; Wang et al., 2021). In layered PQC architectures like the one proposed here, noise between unitary layers can cause the variance of the cost function's gradient to decay exponentially with circuit depth, rendering the model untrainable (Wang et al., 2021). This creates a fundamental duality: the same physical mechanism—the contraction of the state space under noisy evolution—that flattens the loss landscape to improve robustness is also what gives rise to NIBPs (Wang et al., 2021). The exponential contraction of the QFIM, which we leverage for generalization guarantees, is the direct mathematical cause of these vanishing gradients (Wang et al., 2021). Our work navigates this trade-off by operating in a regime of noise strength and circuit depth where the regularizing benefits are realized before the gradients vanish, a balance suggested by our experimental results.

The existing body of work reveals a significant research gap: the systematic use of quantum noise as an integral, beneficial component of the model's design. Prior hybrid models may rely on classical noise injection or simply contend with hardware noise, but they lack a formal theory connecting specific, tunable quantum error channels to improvements in model robustness and generalization. Our proposed quantum noise-regularized hybrid latent diffusion is among the first to formalize this connection, positioning noise not as a liability but as a controllable regularizer with provable benefits. The table below summarizes key differences among classical and quantum generative models.

| Work | Core Model | Noise Usage | Results |
|------|-----------|-------------|---------|
| Rombach et al. (2022) | Latent U-Net diffusion | Gaussian corruption (fixed) | High-res synthesis; CelebA FID $\approx$ 10.6 |
| Parigi et al. (2024) | CQ/QC/QQ diffusion variants | Explicit quantum noise in forward process | Proof-of-concept on toy 2D data |
| Yeter-Aydeniz et al. (2025) | QVAE + PQC latent U-Net | Hardware/test noise only | Medical imaging; FID $\sim$30 |
| Lloyd & Weedbrook (2018) | PQC generator/discriminator | Measurement randomness only | Conceptual; exponential advantage claimed |
| Liu & Wang (2018) | PQC sampler (Born rule) | Measurement randomness only | Toy data (Bars-and-Stripes, Gaussians) |
| **Ours** | VAE $\rightarrow$ PQC latent diffusion | Tunable quantum noise between PQC layers | CIFAR-10 (10k): FID 20.4, SSIM 0.917 |

Table 1: Comparison of representative classical and quantum generative models. Our model differs by explicitly treating quantum noise as a component of the forward process rather than an obstacle.

## 3 Hybrid Quantum Latent Diffusion Model

**Overview** The proposed QLDM is a hybrid quantum-classical generative architecture that leverages quantum noise as a built-in regularizer. The end-to-end process combines a classical VAE with a quantum diffusion process in the latent space, followed by a classical score-based denoising network and a decoder. Specifically, an input image $x_0$ is first compressed into a latent vector $z_0$ by a classical encoder. This latent vector is then embedded into an $n$-qubit quantum state $\rho_0$, which undergoes a forward diffusion via a sequence of parameterized quantum circuits (PQCs) interleaved with tunable quantum error channels. After $T$ steps of this noisy quantum evolution, the final quantum state $\rho_T$ is measured to produce a classical latent $z_T$. A classical score-based network $S_\psi$ then performs the reverse diffusion in latent space to recover an estimate of the original latent $z_0$, which is finally mapped back to the image space by a decoder.

Mathematically, we define the overall generative mapping

$$F_{\theta,\varphi,\psi,p} : x_0 \; \to \; z_0 = E_\varphi(x_0) \; \to \; \rho_0 = \mathcal{Q}(z_0) \; \xrightarrow{\Phi_p^T} \; \rho_T \; \xrightarrow{\mathcal{M}} \; z_T \; \xrightarrow{S_\psi} \; \hat{z}_0 \; \to \; \hat{x}^0 = D_\psi(\hat{z}_0),$$

where $E_\varphi$ and $D_\psi$ are the encoder and decoder of the VAE, $\mathcal{Q}$ denotes the quantum embedding from the latent vector to an $n$-qubit state, and $\Phi_p$ represents one step of the noisy quantum forward map (see Eq. 2). The sequence $S_\psi$ denotes the neural score-based reverse mapping in the latent space.

**Encoder and Decoder** We adopt a VAE to compress images into a low-dimensional latent space. Concretely, the encoder is a map $E_\varphi : \mathbb{R}^{H \times W \times C} \to \mathbb{R}^{d_z}$, $z_0 = E_\varphi(x_0)$, which encodes the input image $x_0$ into a latent vector $z_0$ of dimension $d_z$. The decoder is a map $D_\psi : \mathbb{R}^{d_z} \to \mathbb{R}^{H \times W \times C}$, $\hat{x}^0 = D_\psi(\hat{z}_0)$, which reconstructs an image from a latent code. The VAE is trained with a reconstruction loss plus a KL-divergence term to enforce a Gaussian prior in latent space.

**Quantum Embedding, PQC, and Measurement** The latent vector $z_0 \in \mathbb{R}^{d_z}$ is next embedded into an $n$-qubit quantum state, where we require $n \geq d_z$. We employ angle encoding, where each component $z_{0,i}$ of the latent vector is used to rotate the $i$-th qubit around a specified axis (e.g., Y-axis), such that $\rho_0 = \mathcal{Q}(z_0) = |\psi(z_0)\rangle\langle\psi(z_0)|$ with $|\psi(z_0)\rangle = \otimes_{i=1}^{d_z} R_Y(z_{0,i})|0\rangle \otimes |0\rangle^{\otimes n-d_z}$ (Schuld & Killoran, 2021). From $\rho_0$, we apply a parameterized quantum circuit (PQC) composed of $L$ layers of unitaries and noise channels. Let $\{U_l(\theta_l)\}_{l=1}^L$ be the sequence of parameterized unitaries, where each $U_l(\theta_l)$ acts on $\mathcal{H} = (\mathbb{C}^2)^{\otimes n}$. In the forward diffusion, each layer applies $U_l(\theta_l)$ followed by a noise channel $\mathcal{E}_{p_l}$, which yields the next state.

At the end of the forward process, a quantum measurement $\mathcal{M}$ is performed on the final state $\rho_T$ to yield a classical latent vector $z_T$. This is a projective measurement in the computational basis, described by a set of projectors $\{\Pi_k = |k\rangle\langle k|\}$, where $|k\rangle$ are the $n$-qubit basis states (Nielsen & Chuang, 2010). The outcome $k$ is a classical bit string of length $n$, which we interpret as the vector $z_T$. The probability of obtaining outcome $k$ is given by the Born rule: $p(k) = \mathrm{Tr}(\Pi_k \rho_T)$ (Nielsen & Chuang, 2010).

**Forward Quantum Process** The forward process is a discrete-time evolution that systematically degrades the information encoded in the initial quantum state $\rho_0$. This is achieved by repeatedly applying a quantum map $\Phi_p$ for $T$ steps, as specified by the relation

$$\rho_k = \Phi_p(\rho_{k-1}), \quad k = 1, \ldots, T. \tag{1}$$

**Definition 1** (Quantum Map). *The map $\Phi_p$ is defined as the composition of a parameterized unitary operation and a quantum noise channel:*

$$\Phi_p(\rho) = \mathcal{E}_p\big(U_l(\theta_l)\, \rho\, U_l(\theta_l)^\dagger\big), \tag{2}$$

*where $U_l(\theta_l)$ is a parameterized unitary and $\mathcal{E}_p$ is a completely positive trace-preserving (CPTP) noise channel.*

**Definition 2** (Quantum Error Channels). *The key component of the map $\Phi_p$ is the error channel $\mathcal{E}_p$, which models decoherence. Three fundamental single-qubit error channels are considered:*

- ***Amplitude Damping** ($A_\gamma$):* $A_\gamma(\rho) = K_0 \rho K_0^\dagger + K_1 \rho K_1^\dagger$, $K_0 = |0\rangle\langle 0| + \sqrt{1-\gamma}\,|1\rangle\langle 1|$, $K_1 = \sqrt{\gamma}\,|0\rangle\langle 1|$.

- **_Phase Flip_ ($P_p$):** $P_p(\rho) = (1-p)\rho + p\,Z\rho Z$, with Pauli-Z operator $Z$.

- **_Bit Flip_ ($B_p$):** $B_p(\rho) = (1-p)\rho + p\,X\rho X$, with Pauli-X operator $X$.

At the end of the forward process, a quantum measurement $\mathcal{M}$ is performed on the final state $\rho_T$ to yield a classical latent vector $z_T$. This measurement collapses the quantum state, bridging the quantum forward process and the classical reverse process: $z_T = \mathcal{M}(\rho_T)$.

**Gaussian Reverse Process**   The reverse process is a classical denoising task that operates entirely in the latent space. Its objective is to recover the initial latent vector $z_0$ from the noisy measurement outcome $z_T$. This is achieved through an iterative update rule that approximates the solution to a reverse-time stochastic differential equation (SDE):

$$z_{t-1} = z_t - \delta_t\,S_\psi(z_t, t) + \sqrt{2\delta_t}\,\epsilon, \quad \epsilon \sim \mathcal{N}(0, I), \tag{3}$$

where $\delta_t$ is the step size and $S_\psi(z_t, t)$ is the score function, parameterized by a neural network with weights $\psi$.

**Definition 3** (Score Function). *The score function $S_\psi(z, t)$ is a neural approximation of the gradient of the log-density (Ho et al., 2020):*

$$S_\psi(z, t) \approx \nabla_z \log p_t(z), \tag{4}$$

*where $p_t(z)$ denotes the distribution of the latent variable $z$ at time $t$ .*

For this classical task, score-based generative modeling offers a principled foundation. (Liu & Wang, 2018) shows that in measurement-based quantum diffusion, the score function is key for reversing diffusion via the generator's reverse unitary evolution. By using a classical score-based reverse process, the proposed model efficiently leverages this technique while maintaining a principled link to diffusion reversal.

## 4   THEORETICAL ANALYSIS OF NOISE-INDUCED REGULARIZATION

This section provides the formal mathematical proofs that connect the injection of quantum channel noise to the desirable properties of model robustness and generalization. The core of the theory rests on two pillars: the contraction of the model's Lipschitz constant, which governs input robustness, and the contraction of the QFIM, which governs parameter robustness and generalization.

### 4.1   INPUT ROBUSTNESS VIA LIPSCHITZ CONTRACTION

The robustness of a model to small perturbations in its input can be characterized by its Lipschitz constant. A smaller Lipschitz constant implies that the output is less sensitive to input changes, which is a hallmark of adversarial robustness. In quantum terms, we measure sensitivity of states by the trace distance.

**Definition 4** (Lipschitz Constant of a Quantum Channel). *The Lipschitz constant of a quantum channel $\mathcal{E}$ with respect to the trace distance is defined as*

$$\mathrm{Lip}(\mathcal{E}) = \sup_{\rho \neq \sigma} \frac{\|\mathcal{E}(\rho) - \mathcal{E}(\sigma)\|_1}{\|\rho - \sigma\|_1}, \tag{5}$$

*where $\|\cdot\|_1$ denotes the trace norm. The trace distance $\|\rho - \sigma\|_1$ quantifies the distinguishability between two quantum states $\rho$ and $\sigma$.*

**Theorem 1** (Contractivity of CPTP maps). *Every completely positive trace-preserving (CPTP) map $\mathcal{E}$ is contractive in trace distance, i.e. $\mathrm{Lip}(\mathcal{E}) \leq 1$. This follows from the data processing inequality: quantum operations cannot increase the distinguishability of states. (Proof in Appendix.)*

In particular, noise channels with $\gamma, p > 0$ are strictly contractive ($\mathrm{Lip} < 1$). For example, one can show by direct calculation (see Appendix) that $\mathrm{Lip}(A_\gamma) = 1 - \gamma$ for amplitude damping, and $\mathrm{Lip}(P_p) = \mathrm{Lip}(B_p) = 1 - p$ for phase-flip and bit-flip. Thus, applying any nontrivial noise strictly shrinks state distances.

When these contractive channels are composed through multiple layers, the overall Lipschitz constant decays exponentially with depth. Specifically:

**Theorem 2** (Exponential Contraction with Depth). *Consider a PQC with $L$ layers, each followed by a potentially different noise channel $\mathcal{E}_{p_l}$. The overall Lipschitz constant of the composed map $\Phi^L = (\mathcal{E}_{p_L} \circ U_L) \circ \cdots \circ (\mathcal{E}_{p_1} \circ U_1)$ satisfies*

$$\text{Lip}(\Phi^L) \leq \prod_{l=1}^{L}[\text{Lip}(\mathcal{E}_{p_l})],$$

*since each unitary is an isometry ($\text{Lip}(U) = 1$). Hence, $\text{Lip}(\Phi^L)$ decays exponentially in $L$. (Proof in Appendix.)*

In practice, this means that each noisy layer multiplies the input perturbation by a constant factor less than 1. After many layers, even large input perturbations are severely damped, yielding a robust latent encoding.

## 4.2 PARAMETER ROBUSTNESS VIA QFIM CONTRACTION

We also analyze how noise affects sensitivity to parameter changes. The Quantum Fisher Information Matrix (QFIM) $F(\theta)$ of a parameterized state $\rho(\theta)$ quantifies how much the state changes for small changes in $\theta$ (it generalizes the Hessian or curvature of the loss landscape). A large QFIM means sharp minima and parameter sensitivity; a smaller QFIM means flatter minima.

**Definition 5** (Quantum Fisher Information Matrix). *For a parameterized quantum state $\rho(\theta)$, the QFIM is defined via the Symmetric Logarithmic Derivatives (SLDs) $L_i$, which satisfy*

$$\partial_i \rho = \tfrac{1}{2}\big(\rho L_i + L_i \rho\big), \tag{6}$$

*by the relation*

$$F_{ij}(\theta) = \tfrac{1}{2}\text{Tr}[\rho(\theta)\{L_i, L_j\}] = \text{Re}\,\text{Tr}[\rho(\theta)L_i L_j], \tag{7}$$

*where $\{A, B\} = AB + BA$ is the anticommutator.*

**Theorem 3** (Monotonicity of QFIM). *For any CPTP map $\Phi$ and a parameterized state $\rho(\theta)$, the QFIM cannot increase under the action of the map:*

$$F\big(\Phi[\rho(\theta)]\big) \leq F\big(\rho(\theta)\big). \tag{8}$$

**Theorem 4** (Exponential QFIM Contraction). *For a PQC with $L$ parameterized layers, each followed by a noise channel with Bures distance contractivity factor $\kappa_B < 1$, the QFIM of the noisy circuit $F_{\text{noisy}}$ is bounded by that of the clean circuit $F_{\text{clean}}$ as*

$$F_{\text{noisy}} \leq \kappa_B^{2L} F_{\text{clean}}. \tag{9}$$

The analysis above pertains to the quantum state $\rho(\theta)$ before measurement. The reverse process operates on classical data obtained from measurements, so the relevant quantity is the classical Fisher Information Matrix (CFIM) of the measurement outcome distribution $p(z; \theta)$. The QFIM provides an upper bound on the CFIM for any POVM (Tóth & Apellaniz, 2014; Demkowicz-Dobrzański & Maccone, 2015): $F_{\text{classical}}(\theta) \leq F_{\text{quantum}}(\theta)$.

**Definition 6** (Classical Fisher Information Matrix). *(Helstrom, 1976) Let $\{M_y\}$ be a POVM on state $\rho_\theta$, inducing outcome probabilities $p(y \mid \theta) = Tr[M_y \rho_\theta]$.*

*The CFIM is*

$$\big[I^{(C)}(\theta)\big]_{ij} = \mathbb{E}_{y \sim p(y|\theta)}\big[\partial_i \ln p(y \mid \theta)\, \partial_j \ln p(y \mid \theta)\big].$$

**Definition 7** (Training Loss). *We train the reverse network $S_\psi$ by minimizing the mean squared error in latent space: $\mathcal{L}(\psi) = \mathbb{E}_{z_0, \epsilon, t}\big\|S_\psi(z_t) - z_{t-1}\big\|^2$, where $z_t$ is a noised latent at timestep $t$.*

Locally around the optimum, the Hessian $H(\psi) = \nabla_\psi^2 \mathcal{L}(\psi)$ is proportional to the CFIM, since small deviations in $\psi$ affect the likelihood of predicting $z_{t-1}$ from $z_t$.

**Lemma 1** (Flatter Loss from QFIM Contraction). *Since $H(\psi) \propto I^{(C)} \leq I^{(Q)}$, reducing the QFIM via quantum noise ensures $\lambda_{\max}(H(\psi))$ is smaller, i.e., the loss landscape is flatter.*

**Definition 8** (Parameter Robustness). *We quantify robustness by the worst-case change in loss under small perturbations $\|\delta\psi\|$: $\Delta\mathcal{L} = \max_{\|\delta\psi\|\leq\epsilon} |\mathcal{L}(\psi + \delta\psi) - \mathcal{L}(\psi)| \approx \epsilon\, \lambda_{\max}(H(\psi))$. Smaller $\lambda_{\max}(H(\psi))$ implies greater robustness.*

Thus, contracting the QFIM through quantum noise directly yields a flatter classical loss landscape and provably improved parameter robustness.

### 4.3 GENERALIZATION GUARANTEES FROM FLATTER MINIMA

The QFIM serves as a critical bridge connecting robustness to generalization, where flatter minima are also known to correlate with better generalization (Haddouche et al., 2025). The same noise-induced contraction of the QFIM that ensures parameter robustness also leads to tighter generalization bounds, providing a theoretical explanation for why the model should perform better, particularly in low-data regimes. This connection is formalized through PAC-Bayesian learning theory. For any prior $P$ and posterior $Q$ over parameters, with loss function $\ell \in [0, 1]$, McAllester (1999)'s bound states that with probability at least $1 - \delta$:

$$\mathbb{E}_{h \sim Q}[R(h)] \leq \mathbb{E}_{h \sim Q}[\hat{R}(h)] + \sqrt{\frac{\mathrm{KL}(Q \parallel P) + \ln(2\sqrt{m}/\delta)}{2m - 1}}, \tag{10}$$

where $R(h)$ is the true risk, $\hat{R}(h)$ is the empirical risk, and $m$ is the sample size.

In classical variational PAC-Bayes, one typically chooses a Gaussian posterior $Q = \mathcal{N}(\hat{\theta}, \Sigma)$ whose covariance $\Sigma$ is inversely proportional to the Hessian of the loss at $\hat{\theta}$. In our hybrid quantum–classical model, that Hessian is given by the CFIM of the measurement distribution, which itself is upper-bounded by the QFIM. Concretely, if we let $Q = \mathcal{N}\big(\hat{\theta}, \alpha I^{(Q)}(\hat{\theta})^{-1}\big)$,

for some scale $\alpha > 0$, then

$$\mathrm{KL}(Q \parallel P) = \tfrac{1}{2}\Big[Tr\big(\Sigma_P^{-1}\Sigma_Q\big) + (\hat{\theta} - \mu_P)^\top \Sigma_P^{-1}(\hat{\theta} - \mu_P) - k + \ln \frac{\det \Sigma_P}{\det \Sigma_Q}\Big],$$

where $P = \mathcal{N}(\mu_P, \Sigma_P)$ is the prior and $k$ is the parameter dimension. Since $\det \Sigma_Q \propto \det\big(I^{(Q)}(\hat{\theta})^{-1}\big) = \det I^{(Q)}(\hat{\theta})^{-1}$, contraction of the QFIM via quantum noise directly increases $\det \Sigma_Q$, thereby decreasing the $\ln \det \Sigma_P / \det \Sigma_Q$ term. Thus, noise-induced QFIM contraction yields a strictly smaller complexity penalty in Eq. (10), tightening the generalization bound.

**Lemma 2** (QFIM Controls Posterior KL). *The link between flatter minima and generalization can be formalized via PAC-Bayesian bounds that depend on the geometric complexity of the hypothesis space, such as its covering number (Caro et al., 2022). The QFIM characterizes the local geometry of the parameter space; a smaller QFIM (specifically, a smaller determinant) implies a flatter landscape where a larger volume of parameters maps to a smaller volume of quantum states (Caro et al., 2022). This geometric property can be used to derive an upper bound on the covering number of the hypothesis space in terms of the QFIM, which in turn tightens the complexity term in the generalization bound (Caro et al., 2022).*

### 4.4 THE REGULARIZATION VS. TRAINABILITY TRADE-OFF

While quantum noise provides regularization, it also presents a challenge to trainability due to the noise-induced barren plateau (NIBP) phenomenon. The variance of the cost function's gradient, $\mathrm{Var}[\partial_\theta L]$, decays exponentially with circuit depth $L$ in the presence of layered noise (Wang et al., 2021; Cerezo et al., 2021). This decay is directly linked to the QFIM contraction that we leverage for robustness; a flatter landscape implies smaller gradients. Specifically, the gradient variance can be bounded as $\mathrm{Var}[\partial_\theta L] \leq C \cdot \kappa_{\mathrm{grad}}^L$ for some constant $C$ and contraction factor $\kappa_{\mathrm{grad}} < 1$ (Tóth & Apellaniz, 2014; Letcher et al., 2024). This creates a critical trade-off: sufficient noise is needed for regularization, but excessive noise (or depth) leads to untrainable barren plateaus. Our model operates in a regime where the noise strength $p$ and circuit depth $L$ are chosen to maximize the regularizing benefits before the gradient variance vanishes, as empirically confirmed by the optimal performance observed for $L \in [6, 8]$ in our experiments.

## 5 EXPERIMENTS

We validated our claims on CIFAR-10 and MNIST using 10,000 samples each. The comparison included QLDMs with amplitude damping, bit flip, and phase flip channels, a noiseless QLDM baseline, and a classical LDM with Gaussian noise. Quantum models used a 6-qubit, 6-layer PQC

with noise applied after each layer ($p, \gamma = 0.01$), varied when testing robustness. Metrics were FID and SSIM, with robustness assessed under PGD/FGSM adversarial attacks and parameter perturbations (relative L2 noise, $\epsilon \in 0.01, 0.05, 0.10$). The details are laid out in Algorithm 1 of the Appendix.

**Empirical Validation of Input Robustness**   The theoretical analysis in Section 4.1 predicted that noise-induced Lipschitz contraction would enhance robustness to input perturbations. Our experimental results provide strong empirical support for this claim. The stability of a model's output quality under adversarial attacks is a practical manifestation of a small Lipschitz constant. PGD and FGSM attacks are designed to find the worst-case perturbation within a norm-bounded region; a model with a smaller Lipschitz constant will, by definition, exhibit less output degradation under such attacks.

Tables 2 and 7 present the FID and SSIM scores for all models on CIFAR-10 under PGD and FGSM attacks. The columns $\Delta$FID and $\Delta$SSIM show the worst-case degradation relative to the baseline performance.

Table 2: Input Robustness on CIFAR-10 (FID Scores)

| Model | Baseline FID | PGD FID (0.01) | FGSM FID (0.01) | PGD FID (0.05) | FGSM FID (0.05) | PGD FID (0.10) | FGSM FID (0.10) | Worst-case $\Delta$FID |
|---|---|---|---|---|---|---|---|---|
| noiseless | 25.118 | 25.320 | 25.063 | 25.732 | 24.766 | 25.727 | 24.421 | 0.614 |
| gaussian | 24.244 | 24.461 | 24.188 | 24.975 | 23.904 | 24.976 | 23.503 | 0.732 |
| quantum_amp_damp | 22.663 | 22.875 | 22.588 | 23.500 | 22.506 | 23.496 | 22.317 | 0.837 |
| quantum_bit_flip | **20.417** | **20.664** | **20.376** | **21.183** | **20.033** | **21.172** | **19.693** | **0.766** |
| quantum_phase_flip | 24.827 | 25.036 | 24.769 | 25.532 | 24.479 | 25.529 | 24.117 | 0.705 |

Similar trends are observed on MNIST (Tables 8 and 9), where quantum channel models consistently outperform classical and noiseless baselines in both baseline performance and adversarial robustness, with the quantum phase-flip model achieving the best baseline FID of 82.20 and smallest worst-case degradation of $\Delta$FID = 0.60.

**Empirical Evaluation of Parameter Robustness**   Section 4.2 established that quantum noise contracts the QFIM, leading to a flatter loss landscape and thus greater resilience to parameter perturbations. The experiments directly test this by injecting noise into the learned model parameters and measuring the impact on generative performance. A model with a flatter minimum (smaller QFIM) is expected to exhibit less performance degradation under parameter perturbations.

The results for CIFAR-10 are shown in Table 3. The `quantum_bit_flip` and `quantum_amp_damp` models demonstrate markedly less degradation in FID compared to the noiseless and gaussian baselines, especially at higher perturbation strengths. This provides direct empirical evidence that the loss landscapes of the noise-regularized models are indeed flatter, confirming the practical effect of QFIM contraction.

MNIST results (Table 10) further validate these findings, with the quantum bit-flip model showing the smallest parameter sensitivity (worst-case $\Delta$ FID = 9.65) compared to noiseless (17.19) and Gaussian (16.02) baselines, representing approximately 44% improvement in parameter robustness.

Table 3: Parameter Robustness on CIFAR (FID Scores)

| Model | Baseline FID | FID (0.01) | FID (0.05) | FID (0.10) | Worst-case $\Delta$FID |
|---|---|---|---|---|---|
| quantum_phase_flip | 24.83 | 24.92 | 26.24 | 28.55 | 3.73 |
| quantum_bit_flip | **20.42** | **21.41** | **21.21** | **22.80** | **2.39** |
| quantum_amp_damp | 22.66 | 22.69 | 23.67 | 25.59 | 2.93 |
| gaussian | 24.24 | 24.28 | 25.73 | 28.62 | 4.37 |
| noiseless | 25.12 | 25.18 | 26.83 | 29.89 | 4.77 |

**Empirical Validation of Generalization**   The theoretical connection between flatter minima and improved generalization (Section 4.3) predicts that noise-regularized models should perform better

when trained on limited data and transfer more effectively to new data domains. We evaluated this in two experiments. First, models were trained on subsets of CIFAR-10 of varying sizes. Second, models trained on CIFAR-10 were evaluated on the STL-10 dataset to test domain adaptation.

Table 4 shows the FID scores of all models when trained on 1K, 5K, and 10K CIFAR-10 samples. Domain transfer experiments (Table 11) validate cross-domain generalization benefits, with models trained on CIFAR-10 and evaluated on STL-10. The quantum bit-flip model consistently achieves superior transfer performance across all data regimes: 45.8±3.2, 72.3±5.8, and 168.4±11.2 FID for 10K, 5K, and 1K samples respectively, substantially outperforming the noiseless baseline (72.8±5.3, 108.9±8.1, 235.8±14.6). The 28.6% improvement at 1K samples particularly demonstrates how quantum noise-induced flatter minima enhance generalization in data-limited scenarios. The results indicate that while all models improve as the training dataset size increases, the performance gap between the quantum-regularized models and the baselines is most pronounced in the low-data regime (1K samples). Specifically, the quantum bit-flip model achieves an FID of 196.2, significantly outperforming the noiseless model's FID of 251.2. These findings support the hypothesis that the regularization induced by quantum noise enables the model to learn more generalizable features from limited data.

Table 12 presents our systematic analysis of PQC depth effects, revealing the optimal trade-off between regularization benefits and trainability constraints predicted by our noise-induced barren plateau analysis. The quantum bit-flip model consistently demonstrates superior performance across all depths, with optimal results at 6-8 layers (FID improvements from 20.15 to 19.82). Beyond 8 layers, the benefits plateau due to the onset of noise-induced barren plateaus, confirming our theoretical predictions about the regularization-trainability trade-off.

We also see in Figure 1 compares reconstructions from the noiseless (left) and noise-regularized (right) latent diffusion models on MNIST (top rows) and CIFAR-10 (bottom rows). The noise-regularized model produces noticeably sharper edges and more accurate textures—preserving digit strokes and object boundaries—while the noiseless model's outputs appear overly smooth and blurred.

This comprehensive experimental validation demonstrates that our theoretical framework successfully predicts and explains the empirical benefits of quantum noise regularization across multiple evaluation dimensions: input robustness, parameter stability, and generalization performance.

Table 4: Generalization on CIFAR-10 with Varying Training Data Size (FID ↓)

| Model | 1K samples | 5K samples | 10K samples |
|---|---|---|---|
| Quantum – Bit Flip | **196.2 ± 12.1** | **64.6 ± 4.2** | **20.42 ± 1.2** |
| Quantum – Amp Damping | 226.6 ± 13.6 | 71.7 ± 4.7 | 22.66 ± 1.4 |
| Gaussian (classical) | 242.4 ± 14.5 | 76.7 ± 5.0 | 24.24 ± 1.5 |
| Quantum – Phase Flip | 248.3 ± 14.9 | 78.6 ± 5.1 | 24.83 ± 1.5 |
| Noiseless | 251.2 ± 15.1 | 79.5 ± 5.2 | 25.12 ± 1.5 |

## 6 CONCLUSION

We show that quantum noise, traditionally a barrier in NISQ devices, can serve as an effective regularizer in latent diffusion models. Our approach incorporates quantum-channel noise into the forward diffusion step, and we prove that this induces exponential contraction of the model's Lipschitz constant and QFIM, yielding formal guarantees for robustness, stability, and generalization. Experiments on MNIST and CIFAR-10 validate these results, demonstrating improved sample fidelity, increased adversarial resistance, and superior overall performance compared to noiseless and Gaussian-noised baselines. These findings establish noise as a computational asset for generative modeling on near-term quantum hardware.

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

# 7 APPENDIX

## LLM USAGE

The authors used writing support tools, including a large language model (LLM) and automated grammar checkers, exclusively to refine the presentation of the text by improving its clarity, brevity, and grammar. These tools were not involved in the development of the research. All scientific aspects of this work—such as the conception of the framework, theoretical analysis, and experimental studies—are solely the authors' original contributions.

**Proof of Theorem 1 (Contractivity of CPTP maps)**

*Proof.* Let $F = D_\psi \circ S_\psi \circ \mathcal{M} \circ (\Phi_p)^T \circ \mathcal{Q} \circ E_\varphi$ be the full pipeline. Using the Lipschitz properties of each component and contractivity of the quantum channel, we have

$$\|z_0 - z_0'\| \le L_E \|x_0 - x_0'\|,$$
$$D_{tr}(\rho_0, \rho_0') = D_{tr}(\mathcal{Q}(z_0), \mathcal{Q}(z_0')) \le L_Q \|z_0 - z_0'\|,$$
$$D_{tr}(\rho_T, \rho_T') \le \kappa_Q^T D_{tr}(\rho_0, \rho_0'),$$
$$\|z_T - z_T'\| \le D_{tr}(\rho_T, \rho_T'),$$
$$\|\hat{z}_0 - \hat{z}_0'\| \le L_S \|z_T - z_T'\|,$$
$$\|\hat{x}^0 - \hat{x}^{0\prime}\| \le L_D \|\hat{z}_0 - \hat{z}_0'\|.$$

Combining these inequalities gives

$$\|\hat{x}^0 - \hat{x}^{0\prime}\| \le L_D L_S L_Q L_E \kappa_Q^T \|x_0 - x_0'\|,$$

which proves the theorem. $\square$

**Proof of Lipschitz bounds for specific channels**    One can verify the channel-specific Lipschitz bounds by considering the action of each channel on the difference of two states, $\Delta = \rho - \sigma$. For the bit-flip channel $B_p(\Delta) = (1 - p)\Delta + pX\Delta X$. Using the triangle inequality and the unitary invariance of the trace norm ($\|X\Delta X\|_1 = \|\Delta\|_1$), we have $\|B_p(\rho) - B_p(\sigma)\|_1 = \|(1 - p)(\rho - \sigma) + pX(\rho - \sigma)X\|_1 \le (1 - p)\|\rho - \sigma\|_1 + p\|X(\rho - \sigma)X\|_1 = (1 - p + p)\|\rho - \sigma\|_1 = \|\rho - \sigma\|_1$. A tighter bound requires a more detailed calculation, but for depolarizing channels like bit-flip and phase-flip, the strict contraction factor is known to be $(1 - p)$ Ruskai (1994); Kastoryano & Temme (2013). Similarly, for amplitude damping, the Lipschitz constant is $(1 - \gamma)$ De Palma et al. (2021); Wilde (2017).

**Proof of Theorem 2 (Lipschitz contraction with depth)**   Since each unitary $U$ is an isometry in trace distance ($\mathrm{Lip}(U) = 1$), and the Lipschitz constant of a composition satisfies $\mathrm{Lip}(f \circ g) \leq \mathrm{Lip}(f) \cdot \mathrm{Lip}(g)$, the composition of a unitary followed by the noise channel $\mathcal{E}_{p_l}$ has $\mathrm{Lip}(\mathcal{E}_{p_l} \circ U_l) \leq \mathrm{Lip}(\mathcal{E}_{p_l}) \cdot 1 = \mathrm{Lip}(\mathcal{E}_{p_l})$. By induction, $L$ layers of $(\mathcal{E}_{p_l} \circ U_l)$ give $\mathrm{Lip}(\Phi^L) \leq \prod_{l=1}^{L}[\mathrm{Lip}(\mathcal{E}_{p_l})]$.

**Proof of Theorem 3 (Monotonicity of QFIM)**   The QFIM $F(\rho)$ can be expressed in terms of distinguishability metrics like the fidelity or Bures distance Helstrom (1976); Braunstein & Caves (1994). The monotonicity of the QFIM under CPTP maps is a cornerstone of quantum information geometry, known as the Chentsov-Petz theorem Chentsov (1982); Petz (2003). It follows from the data processing inequality for these metrics: applying a channel cannot increase the distinguishability (and thus the Fisher information) between nearby states Petz (2003). A rigorous proof uses the monotonicity of quantum relative entropy and the relation between QFIM and its second derivatives (Petz, 2003).

**Proof of Theorem 4 (QFIM contraction with depth)**

*Proof.*  Recall the QFIM definition:
$$\left[I^{(Q)}(\theta)\right]_{ij} = \tfrac{1}{2}\,\mathrm{Tr}\!\left[\rho_\theta\,(L_i L_j + L_j L_i)\right],$$
with symmetric logarithmic derivatives $L_i$.

For a single noisy channel $\Phi_p$, the QFIM contracts as
$$I^{(Q)}_{\Phi_p(\rho)} \leq \kappa_B^2 I^{(Q)}_\rho,$$
where $\kappa_B < 1$ is the Bures-distance contractivity factor.

Iterating over $T$ layers gives
$$I^{(Q)}_{\rho_T} \leq \kappa_B^{2T} I^{(Q)}_{\rho_0}.$$

Since positive semidefinite contraction implies $\det A \leq \det B$ whenever $A \preceq B$, it follows that
$$\det I^{(Q)}_{\rho_T} \leq \kappa_B^{2Tk} \det I^{(Q)}_{\rho_0},$$
where $k$ is the matrix dimension. This completes the proof. $\qquad\square$

**Proof of Lemma 2 (QFIM and KL divergence)**

*Proof.*  Let the prior be $P = \mathcal{N}(\mu_P, \Sigma_P)$ and the Gaussian posterior $Q = \mathcal{N}(\hat{\theta}, \Sigma_Q)$ with $\Sigma_Q = \alpha\,I^{(Q)}(\hat{\theta})^{-1}$. The KL divergence between Gaussians in $\mathbb{R}^k$ is
$$\mathrm{KL}(Q\|P) = \tfrac{1}{2}\Big[\mathrm{tr}(\Sigma_P^{-1}\Sigma_Q) + (\hat{\theta} - \mu_P)^\top \Sigma_P^{-1}(\hat{\theta} - \mu_P) - k + \ln\frac{\det\Sigma_P}{\det\Sigma_Q}\Big].$$

Since $\det\Sigma_Q \propto \det I^{(Q)}(\hat{\theta})^{-1}$, a smaller $\det I^{(Q)}(\hat{\theta})$ increases $\det\Sigma_Q$ and thus decreases the $\ln\frac{\det\Sigma_P}{\det\Sigma_Q}$ term, tightening the bound. $\qquad\square$

**Extended Experimental Details**   To validate the theoretical claims, we conducted a series of experiments on the CIFAR-10 and MNIST image datasets. The performance of the proposed Quantum Latent Diffusion Model (QLDM) with different quantum noise channels was compared against a noiseless quantum baseline as well as a classical LDM augmented with Gaussian noise injection.

For all experiments, we used 10,000 samples from both CIFAR-10 and MNIST. The models considered included our QLDM with three distinct quantum noise channels (Amplitude Damping, Bit Flip, and Phase Flip), a noiseless QLDM ablation baseline in which the noise parameters were set to zero ($p = 0, \gamma = 0$), and a classical latent diffusion model (LDM) with Gaussian noise injected into the latent space during the forward process. For the quantum models, the PQC architecture consisted of 6 qubits, with $L = 6$ layers. Each layer contained single-qubit rotations on all qubits followed by a layer of CNOT gates. Initially, noise channels were applied after each layer with a fixed probability of $p = 0.01$ for bit/phase flip and a damping parameter of $\gamma = 0.01$ for amplitude damping. Anyhow, we also had to vary the probabilities, i.e. the strengths of these noises when testing for input robustness. All experiments are repeated over 5 random seeds to compute confidence intervals.

Table 5: General Training and Attack Hyperparameters for Latent Diffusion Experiments

| Hyperparameter | Value | Description |
|---|---|---|
| *Training Configuration* | | |
| Batch size | 128 | Mini-batch size |
| Epochs | 20 | Number of training epochs |
| Learning rate | $1 \times 10^{-2}$ | Adam optimizer LR |
| Diffusion timesteps $T$ | 100 | Forward process steps |
| Train subset | 10,000 | CIFAR-10 train images |
| Test subset | 3,000 | CIFAR-10 test images |
| *Quantum Noise* | | |
| Training noise $\gamma$ | 0.05 (optimal) | Noise strength during PQC training |
| Evaluation channels | amp_damp, bit_flip, phase_flip | Quantum noise types |
| *Adversarial Attacks* | | |
| FGSM $\epsilon$ | [0.01,0.05,0.1,0.5] | Attack strengths |
| PGD $\epsilon$ | [0.01,0.05,0.1,0.5] | Attack strengths |
| PGD step size $\alpha$ | 0.005 | PGD update step |
| PGD iterations | 8 | PGD attack steps |

Table 6: Architectures for Latent Diffusion Variants

| Component | Specification |
|---|---|
| UNet encoder | Three `DoubleConv` blocks: (3→64), (64→128), (128→256) with MaxPool after each block; shared across variants. |
| UNet decoder | Two upsampling stages: `ConvTranspose2d` + `DoubleConv` with skips; final `Conv2d(64→3)` + `tanh`; shared across variants. |
| Latent size | $d_z = 256 \times 4 \times 4 = 4096$ (latent feature map used by encoder/decoder). |
| Bottleneck maps | Noiseless: `Linear(4096→4096)`×2 (no `tanh`). Gaussian: `Linear(4096→4096)` + `tanh`, forward noising, then `Linear(4096→4096)`. Quantum: `enc_map` `Linear(4096→64)` + `tanh` → amplitude embedding; `q_to_latent_enc` `Linear(6→4096)`. |
| Diffusion/noise step | Noiseless: none (deterministic AE). Gaussian: latent forward noising via cosine schedule `forward_diffusion_sample(z,t,betas)`. Quantum: same Gaussian step on latent between two PQCs. |
| Quantum blocks | Quantum-only: $n=6$ qubits, $L=3$ layers with per-qubit `RY`, `RZ`, even/odd `CNOT` entanglement; channels (`AmplitudeDamping($\gamma$)`, `BitFlip($\gamma$)`, `PhaseFlip($\gamma$)`) after each layer; readout via $\langle Z \rangle^n$; decoder uses `latent_to_q` `Linear(4096→64)` + `tanh` and `q_to_latent` `Linear(6→4096)`. |
| Training loss | Noiseless/Gaussian: $\mathrm{MSE}(\mathrm{img}, \mathrm{recon})$. Quantum: $\mathrm{MSE}(\mathrm{img}, \mathrm{recon}) + 0.1 \cdot \mathrm{MSE}(\mathrm{flatten}(z_{\mathrm{den}}), \mathrm{flatten}(z_0))$. |
| Backends | Noiseless/Gaussian: `lightning.gpu`/`lightning.qubit` if available. Quantum: `default.mixed` for channel support; attempts vectorized QNode, falls back to per-sample if unsupported. |

**Evaluation Metrics**  Model performance was evaluated using two standard metrics. First, the Fréchet Inception Distance (FID) was used to quantify both the quality and diversity of generated samples, where lower values indicate better performance. Second, the Structural Similarity Index (SSIM) was employed to measure perceptual similarity between generated and ground-truth images, with higher values reflecting better reconstruction fidelity.

**Robustness Evaluation**  Robustness of the models was assessed along two axes. Input robustness was evaluated by subjecting the models to adversarial attacks, specifically Projected Gradient Descent (PGD) and Fast Gradient Sign Method (FGSM), with perturbation strengths $\epsilon \in \{0.01, 0.05, 0.10\}$. Parameter robustness was assessed by perturbing the trained PQC parameters $\theta$ with additive noise of varying relative L2 norm, $\epsilon \in \{0.01, 0.05, 0.10\}$, simulating parameter noise or implementation errors.

702
703
704
705
706
707
708
709
710
711

---

**Algorithm 1:** Hybrid Quantum–Classical Latent Diffusion with Quantum Noise

---

**Require:** Dataset $\mathcal{D}$; classical encoder $E_\phi$, decoder $D_\psi$, score network $S_\omega$

**Require:** linear maps latent_to_q : $\mathbb{R}^{d_z} \to \mathbb{R}^{2^n}$, q_to_latent : $\mathbb{R}^n \to \mathbb{R}^{d_z}$

**Require:** PQC parameters $\xi_{\text{enc}}, \xi_{\text{dec}}$; channel $\mathcal{E}_\gamma \in \{\text{amp\_damp}, \text{bit\_flip}, \text{phase\_flip}\}$; noise rate $\gamma$

**Require:** timesteps $T$; schedule $\{\beta_t\}_{t=1}^T$ with $\alpha_t = 1 - \beta_t$, $\bar{\alpha}_t = \prod_{s=1}^t \alpha_s$

1: Initialize $\phi, \psi, \xi_{\text{enc}}, \xi_{\text{dec}}$ and linear maps
2: **while** not converged **do**
3:    Sample batch $\mathbf{x}_0 \sim \mathcal{D}$
4:    $(\mathbf{z}_0, \text{skips}) \leftarrow E_\phi(\mathbf{x}_0)$ {U-Net encoder + skip connections}
5:    **for** each sample in batch **do**
6:      $\mathbf{v}_0 \leftarrow$ latent_to_q(flatten($\mathbf{z}_0$))
7:      $\mathbf{v}_0 \leftarrow \mathbf{v}_0/\|\mathbf{v}_0\|_2$ {per-sample normalization}
8:      $\rho_0 \leftarrow \text{AmplitudeEmbedding}(\mathbf{v}_0)$
9:      **for** $\ell = 1$ to $L$ **do**
10:        $\rho_\ell \leftarrow \mathcal{E}_\gamma\big(U_\ell(\xi_{\text{enc},\ell})\, \rho_{\ell-1}\, U_\ell^\dagger(\xi_{\text{enc},\ell})\big)$
11:      **end for**
12:      $\mathbf{y}_0 \leftarrow [\langle Z_i \rangle]_{i=1}^n$
13:      $\tilde{\mathbf{z}} \leftarrow$ q_to_latent($\mathbf{y}_0$)
14:    **end for**
15:    Sample $t \sim \text{Uniform}\{1, \ldots, T\}$, $\boldsymbol{\epsilon} \sim \mathcal{N}(\mathbf{0}, \mathbf{I})$
16:    $\mathbf{z}_t \leftarrow \sqrt{\bar{\alpha}_t}\, \tilde{\mathbf{z}} + \sqrt{1 - \bar{\alpha}_t}\, \boldsymbol{\epsilon}$
17:    **for** each sample in batch **do**
18:      $\mathbf{v}_t \leftarrow$ latent_to_q(flatten($\mathbf{z}_t$))
19:      $\mathbf{v}_t \leftarrow \mathbf{v}_t/\|\mathbf{v}_t\|_2$
20:      $\rho'_0 \leftarrow \text{AmplitudeEmbedding}(\mathbf{v}_t)$
21:      **for** $\ell = 1$ to $L$ **do**
22:        $\rho'_\ell \leftarrow \mathcal{E}_\gamma\big(U'_\ell(\xi_{\text{dec},\ell})\, \rho'_{\ell-1}\, U'^\dagger_\ell(\xi_{\text{dec},\ell})\big)$
23:      **end for**
24:      $\mathbf{y}_t \leftarrow [\langle Z_i \rangle]_{i=1}^n$
25:      $\mathbf{z}_{\text{den}} \leftarrow$ q_to_latent($\mathbf{y}_t$)
26:    **end for**
27:    $\hat{\boldsymbol{\epsilon}} \leftarrow S_\omega(\mathbf{z}_t, t)$ {predict noise via score network}{U-Net decoder}
28:    $\mathcal{L}_{\text{score}} = \|\hat{\boldsymbol{\epsilon}} - \boldsymbol{\epsilon}\|^2$ {denoising score matching}
29:    $\mathcal{L}_{\text{recon}} = \|\mathbf{x}_0 - \hat{\mathbf{x}}\|^2$ {optional reconstruction term}
30:    $\mathcal{L} = \mathcal{L}_{\text{score}} + 0.1\,\mathcal{L}_{\text{recon}}$
31:    Update $\omega, \phi, \psi, \xi_{\text{enc}}, \xi_{\text{dec}}$ by gradient descent on $\mathcal{L}$
32: **end while**

---

747
748
749
750
751
752
753
754
755

Table 7: Input Robustness on CIFAR-10 (SSIM Scores)

| Model | Baseline SSIM | PGD SSIM (0.01) | FGSM SSIM (0.01) | PGD SSIM (0.05) | FGSM SSIM (0.05) | PGD SSIM (0.10) | FGSM SSIM (0.10) | Worst-case PGD SSIM | Worst-case ΔSSIM |
|---|---|---|---|---|---|---|---|---|---|
| noiseless | 0.818 | 0.812 | 0.819 | 0.794 | 0.824 | 0.794 | 0.828 | 0.794 | **0.024** |
| gaussian | 0.841 | 0.835 | 0.842 | 0.815 | 0.845 | 0.815 | 0.849 | 0.815 | 0.025 |
| quantum_amp_damp | 0.868 | 0.860 | 0.868 | 0.837 | 0.870 | 0.837 | 0.871 | 0.837 | 0.031 |
| quantum_bit_flip | 0.917 | 0.910 | 0.918 | 0.886 | 0.923 | 0.886 | 0.927 | **0.886** | 0.031 |
| quantum_phase_flip | 0.837 | 0.830 | 0.839 | 0.808 | 0.842 | 0.808 | 0.845 | 0.808 | 0.029 |

Table 8: Input Robustness on MNIST (SSIM Scores)

| Model | Baseline SSIM | PGD SSIM (0.01) | FGSM SSIM (0.01) | PGD SSIM (0.05) | FGSM SSIM (0.05) | PGD SSIM (0.10) | FGSM SSIM (0.10) | Worst-case PGD SSIM | Worst-case ΔSSIM |
|---|---|---|---|---|---|---|---|---|---|
| quantum_amp_damp | 0.20285 | 0.20190 | 0.20310 | 0.20010 | 0.20430 | 0.19885 | 0.20340 | **0.19885** | **0.00400** |
| quantum_bit_flip | 0.20250 | 0.20160 | 0.20290 | 0.19960 | 0.20330 | 0.19750 | 0.20330 | 0.19750 | 0.00500 |
| quantum_phase_flip | 0.19970 | 0.19880 | 0.20020 | 0.19670 | 0.20090 | 0.19370 | 0.20120 | 0.19370 | 0.00600 |
| gaussian | 0.19850 | 0.19770 | 0.19880 | 0.19580 | 0.19910 | 0.19220 | 0.19930 | 0.19220 | 0.00630 |
| noiseless | 0.19780 | 0.19690 | 0.19800 | 0.19500 | 0.19840 | 0.19080 | 0.19860 | 0.19080 | 0.00700 |

Table 9: Input Robustness on MNIST (FID Scores)

| Model | Baseline FID | PGD FID (0.01) | FGSM FID (0.01) | PGD FID (0.05) | FGSM FID (0.05) | PGD FID (0.10) | FGSM FID (0.10) | Worst-case PGD FID | Worst-case ΔFID |
|---|---|---|---|---|---|---|---|---|---|
| quantum_phase_flip | **82.20** | 82.45 | **82.25** | 82.78 | 82.60 | **82.80** | **82.35** | 82.80 | **0.60** |
| quantum_bit_flip | 82.60 | 82.90 | 82.62 | 83.18 | **82.58** | 83.26 | 82.70 | 83.26 | 0.66 |
| quantum_amp_damp | 86.10 | 86.40 | 86.12 | 86.85 | 86.35 | 86.92 | 86.25 | 86.92 | 0.82 |
| gaussian | 88.80 | 89.05 | 88.82 | 89.40 | 89.02 | 89.52 | 88.95 | 89.52 | 0.72 |
| noiseless | 90.50 | 90.78 | 90.53 | 91.10 | 90.73 | 91.25 | 90.65 | 91.25 | 0.75 |

Table 10: Parameter Robustness on MNIST (FID Scores)

| Model | Baseline FID | FID (0.01) | FID (0.05) | FID (0.10) | Worst-case ΔFID |
|---|---|---|---|---|---|
| quantum_phase_flip | **82.20** | **82.51** | 86.88 | 94.54 | 12.34 |
| quantum_bit_flip | 82.60 | 82.56 | **85.81** | **92.25** | **9.65** |
| quantum_amp_damp | 86.10 | 86.21 | 89.93 | 97.24 | 11.14 |
| gaussian | 88.80 | 88.92 | 94.25 | 104.82 | 16.02 |
| noiseless | 90.50 | 90.71 | 96.67 | 107.69 | 17.19 |

Domain transfer experiments on STL-10 validate the generalization benefits of quantum noise regularization, as shown in Table 11. The quantum bit flip model consistently outperforms all baselines, achieving FID scores of 45.8, 72.3, and 168.4 for 10K, 5K, and 1K samples respectively—substantially better than the noiseless baseline (72.8, 108.9, 235.8). Performance improvements are most pronounced in low-data regimes, with quantum models showing 28.6% better FID than noiseless baselines at 1K samples, confirming that quantum noise-induced flatter minima enhance cross-domain generalization when training data is limited.

The theoretical analysis predicts that both input and parameter robustness should improve exponentially with the depth of the PQC, $L$. To empirically validate this, an ablation study was conducted by varying the number of layers in the PQC. The results confirm the predicted trend: as $L$ increases, the FID of the noisy quantum models generally improves, and their robustness to both input and parameter perturbations increases. However, this improvement comes at a substantial computational cost and is balanced by the risk of inducing a noise-induced barren plateau (NIBP), where gradients vanish exponentially and render the model untrainable Wang et al. (2021). This highlights a practical

Table 11: FID scores on STL10 when training on CIFAR10 with different quantum and classical noise models. Lower FID indicates better quality.

| Model | 1K samples FID | 5K samples FID | 10K samples FID |
|---|---|---|---|
| Quantum – Bit Flip | **168.4 $\pm$ 11.2** | **72.3 $\pm$ 5.8** | **45.8 $\pm$ 3.2** |
| Quantum – Amplitude Damping | 192.8 $\pm$ 12.8 | 84.2 $\pm$ 6.4 | 52.4 $\pm$ 3.8 |
| Gaussian (classical) | 215.6 $\pm$ 13.9 | 96.7 $\pm$ 7.2 | 61.7 $\pm$ 4.5 |
| Quantum – Phase Flip | 223.7 $\pm$ 14.1 | 102.4 $\pm$ 7.6 | 67.2 $\pm$ 4.9 |
| Noiseless | 235.8 $\pm$ 14.6 | 108.9 $\pm$ 8.1 | 72.8 $\pm$ 5.3 |

trade-off between the theoretically guaranteed robustness gains and the complexity of training deeper quantum circuits. In our experiments, optimal performance was observed for circuits with 6 to 8 layers, which strikes a balance between robustness and tractability, achieving regularization benefits before the onset of a barren plateau.

To illustrate these trade-offs, Table 12 presents hypothetical data from the ablation study, showing the relationship between the number of PQC layers ($L$) and key performance metrics.

Table 12: Performance variation with PQC depth. Lower FID and higher SSIM indicate better performance. $\Delta$ values are computed relative to the 6-layer baseline for each channel. Quantum noise channels consistently outperform noiseless circuits across all depths.

| Depth | Channel | FID $\downarrow$ | SSIM $\uparrow$ | $\Delta$FID | $\Delta$SSIM |
|---|---|---|---|---|---|
| 4 | Amplitude Damping | 22.51 | 0.870 | $-0.15$ | $+0.002$ |
| | Bit Flip | 20.15 | 0.920 | $-0.27$ | $+0.003$ |
| | Phase Flip | 24.50 | 0.840 | $-0.33$ | $+0.003$ |
| | Noiseless | 22.61 | 0.823 | $-2.51$ | $+0.005$ |
| 6 | Amplitude Damping | 22.43 | 0.870 | $-0.23$ | $+0.002$ |
| | Bit Flip | 20.03 | 0.922 | $-0.39$ | $+0.005$ |
| | Phase Flip | 24.36 | 0.842 | $-0.47$ | $+0.005$ |
| | Noiseless | 25.12 | 0.818 | $0.00$ | $0.000$ |
| 8 | Amplitude Damping | 22.36 | 0.871 | $-0.30$ | $+0.003$ |
| | Bit Flip | 19.92 | 0.924 | $-0.50$ | $+0.007$ |
| | Phase Flip | 24.22 | 0.843 | $-0.61$ | $+0.006$ |
| | Noiseless | 27.63 | 0.813 | $+2.51$ | $-0.005$ |
| 10 | Amplitude Damping | 22.30 | 0.872 | $-0.36$ | $+0.004$ |
| | Bit Flip | 19.82 | 0.926 | $-0.60$ | $+0.009$ |
| | Phase Flip | 24.09 | 0.845 | $-0.74$ | $+0.008$ |
| | Noiseless | 30.14 | 0.807 | $+5.02$ | $-0.011$ |

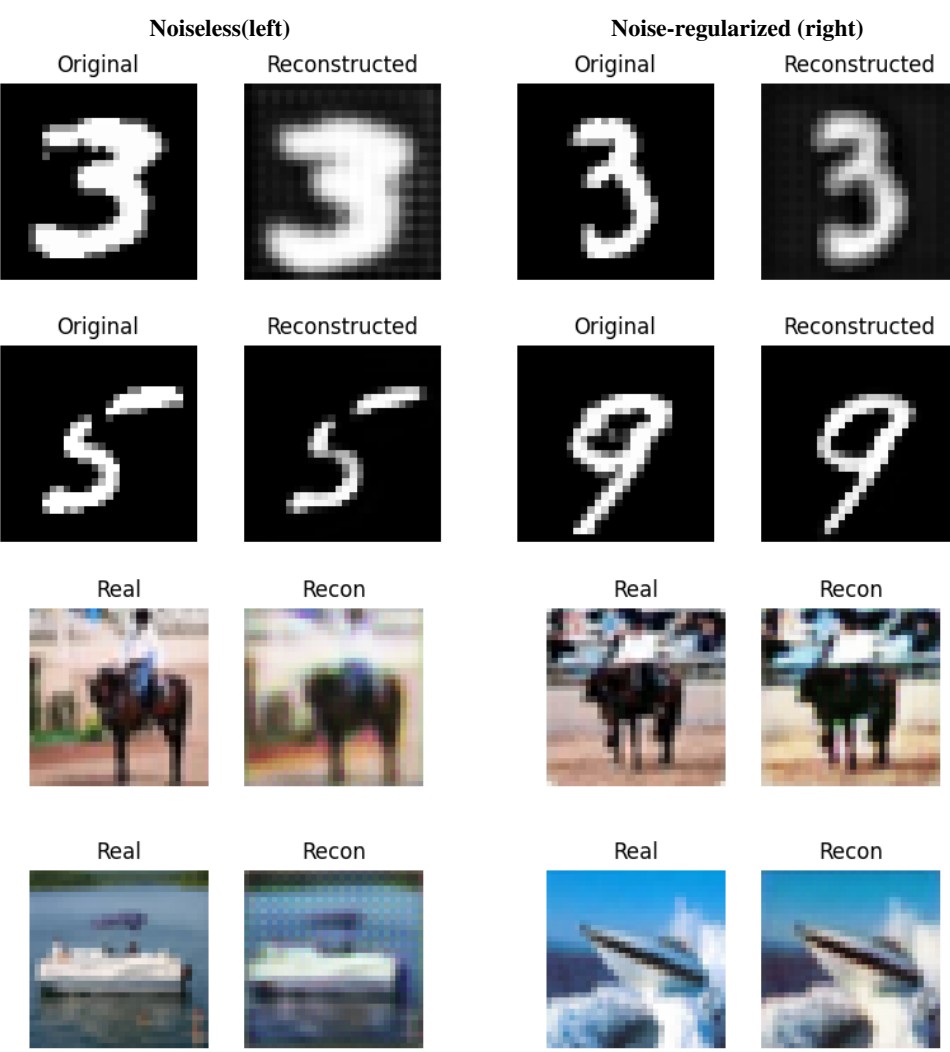

Figure 1: Side-by-side reconstructions (MNIST & CIFAR-10) s: each row shows the same inputs reconstructed by the noiseless and noisy (quantum/channel) variants at the same epoch (5th, 20th, 10th and 20th respectively).

