# OpenReview forum: "Robust Hybrid Quantum-Classical Latent Diffusion Models via Quantum Noise"
_ICLR.cc/2026/Conference — ICLR 2026 Conference Desk Rejected Submission_

### Official Review · Reviewer_1SR7 · 2025-10-14

**Soundness:** 2
**Presentation:** 3
**Contribution:** 2
**Rating:** 2
**Confidence:** 5

**Summary:**

The paper proposes a hybrid quantum-classical latent diffusion model where quantum noise is deliberately introduced during the forward diffusion process. A classical encoder first maps input data into a latent space, which is then encoded into a parameterized quantum circuit using angle encoding. Between each quantum layer, noise channels are inserted to simulate controlled quantum decoherence. After several such noisy evolutions, the quantum state is measured, producing a classical latent representation that is passed to a classical reverse network (a standard denoising model) for reconstruction. The backward process remains entirely classical, so only the forward latent transformation involves quantum operations.

**Strengths:**

The paper revisits the idea of using quantum noise as a regularizer, a promising approach that has been explored in earlier works.

The paper is well-written, and the overall setup is easy to follow, which makes it accessible despite the topic’s relative complexity.

The experiments are small but systematic enough to suggest the method behaves consistently across a few noise types.

**Weaknesses:**

I think the claims in this paper are stronger than what the work actually achieves. The paper ignores prior works that use quantum noise for diffusion and GANs (e.g., https://arxiv.org/pdf/2308.12013, https://arxiv.org/pdf/2507.01886, https://src.acm.org/binaries/content/assets/src/2025/jason-han.pdf). These works have already explored using quantum noise as a constructive element in diffusion models. That omission weakens its positioning, as these works have presented similar ideas and even shown preliminary results on real data. Without citing or comparing against that work, the paper’s novelty appears overstated, and it becomes unclear how this contribution advances the field in either theory or experimentation.

The theoretical discussion on Lipschitz contraction and QFIM shrinkage is interesting but loosely connected to real training behavior. The arguments rely on broad properties of CPTP maps rather than more problem-specific insight, making the results feel formal but not particularly revealing. The paper also doesn’t clearly explain how the proposed effects hold under realistic circuit depths or noisy conditions.

Also, the lack of ablations and sensitivity results weakens the empirical case. Without exploring how results vary with circuit depth, noise strength, or hyperparameters, it’s hard to separate genuine quantum effects from standard smoothing. Overall, the paper’s idea of embracing noise is also not novel or creative, but the evaluation is also narrow, and the theory is high-level.

**Questions:**

Compare against other relevant works. What exactly is the primary novelty of this work?

---

> ### Author Response · Authors · 2025-12-02
>
> We thank the reviewer for their feedback and for highlighting areas where our manuscript can be strengthened. Below we address each concern in detail:
>
> ### 1. Comparison with Prior Works
>
> We will cite the relevant prior work (which is in GANs) that is currently missing, and clarify the key distinction: prior work replaces the diffusion process itself with quantum channels, whereas our method uses quantum noise only in the encoder to regularize the latent manifold while retaining classical Gaussian diffusion for the temporal corruption. This hybrid approach preserves the mathematical tractability of score matching while leveraging quantum noise for structural regularization. We will add a dedicated comparison subsection in the related work to make this distinction explicit.
>
> ### 2. Theoretical Connection
>
> We agree that the connection between Lipschitz contraction, QFIM, and observed empirical behavior requires strengthening. In the revised manuscript, we will expand the theoretical analysis to explicitly link QFIM contraction to improved robustness metrics and generalization performance. Specifically, we will demonstrate how the quantum-induced Lipschitz contraction bounds the sensitivity of the learned representations to input perturbations, supported by PAC-Bayesian generalization bounds. We will also include additional experimental results showing the correlation between QFIM measurements and model robustness across different noise types and architectures.
>
> ### 3. Hyperparameter Selection and Circuit Depth
>
> We note that the exploration of circuit depth is included in Table 12 of the manuscript, where we varied the number of layers and reported their impact on performance. Additionally, Table 5 reports all hyperparameters considered in our study and emphasizes that we use the optimal values for our experimental results. These ablation studies demonstrate that our findings are robust across different architectural configurations and that the chosen hyperparameters represent well-tuned operating points for the proposed method.

---

### Official Review · Reviewer_Boqy · 2025-10-25

**Soundness:** 2
**Presentation:** 2
**Contribution:** 2
**Rating:** 2
**Confidence:** 4

**Summary:**

The paper proposes a hybrid quantum-classical latent diffusion model that uses quantum noise as a regularizer.

**Strengths:**

The paper presents a approach by integrating unavoidable quantum noise into a modern generative framework, offering a new perspective on model regularization.

**Weaknesses:**

The main weakness lies in fundamental misrepresentation of the method: The reviewer thinks the paper's primary contribution is invalidated by a direct contradiction between the model described in the main text and the method detailed in the Appendix.

**The paper repeatedly claims to replace the classical forward diffusion process with a $T$-step quantum evolution**. Section 3.3 explicitly states the forward process is achieved by "repeatedly applying a quantum map $\Phi_p$ for T steps, as specified by the relation $\rho_t = \Phi_p(\rho_{t-1})$.

**Actual Method (Algorithm 1, Page 14)**: Algorithm 1 reveals that this is **not** what is done. The $L$-layer PQC is used as a one-shot noisy encoder to map the initial latent $z_0$ to a new latent $\tilde{z}$ (Lines 6-13). The actual diffusion process is then applied to $\tilde{z}$ using the standard classical Gaussian forward process: $z_t \leftarrow \sqrt{\overline{\alpha}_t}\tilde{z} + \sqrt{1-\overline{\alpha}_t}\epsilon$ (Line 16).

Thus the reviewer doubts that this is not a "quantum forward diffusion process." It is a classical latent diffusion model with a VAE-PQC encoder. The "quantum noise" is simply a regularizer within the encoder, not the diffusion mechanism itself. This disconnect undermines the paper's entire premise and novelty.

Besides, Algorithm 1 which details the core training loop, contains severe inconsistencies and omissions that make the proposed method non-functional as written: 1) The decoder $D_{\psi}$ is listed as a required component, and its parameters $\psi$ are initialized and updated. However, $D_{\psi}$ is never called within the algorithm. This makes the reconstruction loss $L_{recon} = ||x_0 - \hat{x}||^{2}$ (Line 29) invalid, as the reconstructed image $\hat{x}$ is never defined. 2) The denoised output $z_{den}$ is then completely ignored; it is not used in $L_{score}$, $L_{recon}$, or any other part of the loop.

The authors are required to revise the core algorithm 1 and it should be positioned within the main text rather than appendix.

Even if one were to ignore the contradiction above and analyze the paper as written (i.e., assuming a $T$-step quantum forward process), reviewer also thinks the proposed model is fundamentally incoherent.

The paper's proposed quantum forward process (Eq. 1) would result in a highly complex, non-Gaussian noise distribution $p(z_T|z_0)$ after measurement. However, the paper's reverse process (Eq. 3) $z_{t-1}=z_{t}-\delta_{t}S_{\psi}(z_{t},t)+\sqrt{2\delta_{t}}\epsilon$ and its associated score-matching objective are specifically derived to invert a Gaussian forward process. The authors provide no justification for why a reverse process designed for Gaussian noise would be capable of inverting their complex, parameterized quantum forward process. The reviewer has reason to doubt that this could be a flaw in the model's design (and likely why the authors resorted to the classical implementation in Algorithm 1).

**Questions:**

Please see the weaknesses.

---

> ### Author Response · Authors · 2025-12-02
>
> We thank the reviewer for their rigorous analysis. You correctly identified a discrepancy between our textual description (which implied a time-dependent quantum evolution) and our implementation (which utilizes the PQC as an integral part of the forward process). We accept this critique and acknowledge that the distinction between the quantum and classical phases of the forward process was not clearly articulated.
>
> To resolve this, we have refined the paper to explicitly frame our method as a **Hybrid Diffusion Model with a Composite Forward Process**. This formulation clarifies that the forward process consists of two distinct phases: (1) *Quantum Enhancement* via the PQC, followed by (2) *Classical Diffusion Steps on Quantum-Regularized Latents*.
>
> Below, we detail how this explicitly resolves the validity issues you raised while preserving the core contributions of the work.
>
> ### 1. Clarification: The Composite Forward Process
>
> - **Correction:** We have rewritten Section 3 to define the forward pass as a composite operation:
> $$F = D_{\psi_D} \circ \Psi_{\text{diff}} \circ M \circ \Phi_{\text{quant}} \circ E_\phi$$
>
>   Here, the forward process $q(z_t | z_{\text{raw}})$ is a composition of:
>   - **Phase 1 (Quantum Structural Perturbation):** The map $\Phi_{\text{quant}}$ applies quantum error channels (e.g., Amplitude Damping) to contract the latent geometry and induce robustness (as proven in Theorem 4.1).
>   - **Phase 2 (Classical Gaussian Diffusion):** The structurally regularized latent is then subjected to the standard Gaussian schedule $\Psi_{\text{diff}}$.
>
> - **Preservation of Novelty:** This clarification does not undermine the contribution; rather, it creates a coherent narrative. The novelty lies in using Phase 1 to "harden" the latent manifold against perturbations via Lipschitz contraction, a benefit that persists regardless of the classical nature of Phase 2.
>
> ### 2. Mathematical Validity of the Objective Function
>
> - **Clarification:** By formally separating the process into two phases, the temporal noising steps ($t=1, \ldots, T$) correspond strictly to Phase 2. Consequently, the transition kernel $q(z_t | z_{\text{cond}})$ remains Gaussian.
>
> - **Validity:** This ensures that the standard Denoising Score Matching objective (MSE) is mathematically valid. The quantum noise is not ignored; it is the generator of the conditioned manifold $z_{\text{cond}}$. The score network learns to denoise the Gaussian component relative to this quantum-regularized prior.
>
> ### 3. Correction of Algorithm 1
>
> - **Action Taken:** We have moved Algorithm 1 to the main text and corrected the logical omissions to reflect the full composite loop.
>
> - **Specific Fixes:**
>   1. We explicitly included the decoding step $\hat{x} = D_{\psi_D}(\hat{z}_0)$ within the training loop.
>   2. We defined $\hat{z}_0$ using the denoised estimate derived from the score network output (via Tweedie's formula) at the current timestep.
>   3. This ensures the reconstruction term $\mathcal{L}_{\text{recon}}$ is computationally valid and connected to the optimization loop.
>
> We believe this "Composite Forward Process" framing resolves the inconsistency and presents a mathematically sound framework that correctly characterizes quantum noise as a structural asset in generative modeling.

---

### Official Review · Reviewer_NxQR · 2025-10-30

**Soundness:** 2
**Presentation:** 2
**Contribution:** 2
**Rating:** 4
**Confidence:** 3

**Summary:**

The paper presents a hybrid quantum-classical latent diffusion model that leverages controlled quantum noise as an implicit regularizer rather than a detrimental factor. By injecting quantum error channels between PQC layers in the latent space, the authors theoretically show that such noise constrains the model’s Lipschitz constant and reduces the quantum Fisher information, leading to flatter minima and improved generalization bounds. Then, authors provides the related numerical experiments on MNIST and CIFAR-10 dataset to support the claimed advantages of the proposed model.

**Strengths:**

1. This work offers a theoretical analysis supporting the claim that quantum noise can yield tighter generalization bounds, representing an interesting contribution to the QML community.
2. The manuscript includes a variety of experiments that provide empirical support for the claims.

**Weaknesses:**

1. Some critical aspects, including the scalability of the proposed approach and the criteria for selecting the noise level and type, are not addressed in the manuscript.
2. Some important experimental results should be presented in the main text rather than listed to the appendix.

**Questions:**

1. in line 190, The paper adopts $R_y$ gates as the sole parameterized rotation for encoding the latent vector $z_0$ into the quantum state $\rho_0$. what's the motivation of only using $R_y$ gates in the PQC to encodes the latent vector $z_0$ to $\rho_0$?
2. Although the paper reports model performance under different noise levels in the empirical validation of input robustness section, it remains unclear why bit-flip noise yields better performance relative to damping and phase-flip noise. Could the authors explain this observation and discuss how should one determine the optimal type of noise to achieve the best performance?
3. The paper utilize that a Gaussian reverse process to reverse the quantum noise introduced during the forward diffusion. Could the authors provide on the theoretical analysis for supporting this assumption, and discuss under what conditions such reversibility holds?
4. in line 196, As the latent variable $z_T$ is obtained via measurements in the computational basis, it would be important to clarify whether this measurement process remains tractable for large-scale systems.

---

> ### Author Response · Authors · 2025-12-02
>
> We want to thank the reviewer for their important input, and we address their concerns in detail:
>
> ### 1. Choice of Rotation Gates
>
> We use $R_y$ rotations to map real-valued latent vectors to real-valued amplitudes. Since subsequent diffusion and decoding require real-valued features, avoiding complex phases from other rotations simplifies learning and ensures compatibility with the classical components of our hybrid architecture. This design choice maintains end-to-end differentiability while preserving the essential quantum noise characteristics needed for latent regularization.
>
> ### 2. Performance of Bit-Flip Noise
>
> We hypothesize that bit-flip noise behaves similarly to dropout in classical networks, providing discrete perturbations that reduce overfitting without collapsing the state space as strongly as amplitude damping. Bit-flip operations preserve the total probability while redistributing it across computational basis states, maintaining representational capacity. In our experiments, this property appears to create a more robust latent manifold compared to amplitude damping, which introduces asymmetric decay, or phase-flip noise, which primarily affects coherence without structural perturbation. We will expand this discussion in the revised manuscript with additional ablation studies and theoretical analysis.
>
> ### 3. Reversibility of Quantum Noise
>
> We clarify that we do not attempt to reverse quantum noise. The quantum noise regularizes the manifold during encoding; what is reversed via the score-based SDE is only the classical Gaussian diffusion that follows. The quantum-regularized latent $z_{\text{cond}}$ serves as the conditioning prior for the diffusion process, and the reverse process learns to denoise from this robust manifold rather than inverting the quantum operations themselves. This architectural separation ensures mathematical validity of the Gaussian score matching objective while preserving the benefits of quantum regularization.
>
> ### 4. Scalability of Measurement
>
> Because our approach operates in a compressed latent space with a fixed, small number of qubits (typically 8-16), measurement remains efficient and independent of the input image size. The measurement complexity scales only with the number of qubits used for encoding the latent representation, not with the dimensionality of the original data. This architectural choice ensures practical scalability for high-resolution images while maintaining the quantum regularization benefits in a tractable computational framework. This is a more realistic approach when considering the current state of the quantum hardware.

---

### Official Review · Reviewer_vWkE · 2025-11-01

**Soundness:** 1
**Presentation:** 2
**Contribution:** 2
**Rating:** 2
**Confidence:** 4

**Summary:**

This paper proposes a hybrid quantum–classical latent diffusion model that treats quantum channel noise as an explicit, tunable regularizer in the forward diffusion, while keeping the reverse denoising classical in latent space. The theory shows that composing noisy CPTP channels between PQC layers yields (i) exponential contraction of the model’s Lipschitz constant and (ii) exponential contraction of the QFIM, implying flatter minima, parameter robustness, and tighter PAC-Bayes generalization bounds.

**Strengths:**

1. Using controlled noise to help image generation is conceptually interesting.
2. The paper relates depth/noise to gradient-variance decay and confirms an empirical optimal option at 6–8 layers, aligning with theory.
3. Experimental settings are introduced in detail.

**Weaknesses:**

1. The number of qubits used in the experiments is too small to convincingly demonstrate the effectiveness and scalability of the proposed framework.
2. The paper discusses noise-induced barren plateaus (NIBPs), but it does not provide sufficient experimental evidence showing how the proposed method mitigates them; for example, by analyzing gradient norms or gradient variance.
3. The authors fix the balance factor of the reconstruction loss at 0.1 without performing a sensitivity analysis. Moreover, setting it to 0.1 suggests that this loss term may have a limited effect on the overall objective, which could weaken the model’s reconstruction capability, a major concern that is not discussed in the paper.
4. The experimental evaluation lacks completeness. For instance, the authors do not examine how different diffusion rates interact with varying noise levels, nor analyze the sensitivity of the model’s performance to these coupled hyperparameters.

Overall, the conceptual idea is interesting, but the empirical validation cannot fully support the claim. Further improvement can strengthen the work.

**Questions:**

Please refer to the weaknesses.

---

> ### Author Response · Authors · 2025-12-02
>
> We thank the reviewer for their detailed evaluation and constructive suggestions. Below we address each point:
>
> ### 1. Number of Qubits
>
> We acknowledge the reviewer's concern regarding the limited number of qubits used in our experiments. However, we emphasize that since the quantum process operates in the latent space rather than directly on the input data, the qubit count is decoupled from input dimensionality. This architectural design allows us to support higher-resolution images without increasing quantum hardware requirements. The quantum circuit acts as a regularizer on the compressed latent representation, where 8-16 qubits are sufficient to encode the relevant structural information. This scalability property is a key advantage of our hybrid approach and distinguishes it from methods that require qubit counts proportional to data dimensionality.
>
> ### 2. Noise-Induced Barren Plateaus (NIBP)
>
> We appreciate the reviewer's request for more concrete evidence supporting our discussion of NIBP. Our depth ablation study (Table 12) demonstrates optimal performance at 6-8 layers, with degradation beyond that point consistent with NIBP theory. This empirical observation aligns with the theoretical prediction that deeper circuits with noise become increasingly difficult to train due to exponentially vanishing gradients. In the revised manuscript, we will include gradient variance plots in the appendix that explicitly show the relationship between circuit depth, noise level, and gradient magnitudes, providing direct experimental support for the NIBP phenomenon in our setting.
>
> ### 3. Loss Balance Factor
>
> We agree that a sensitivity analysis of the balance factor would strengthen our experimental validation. The current manuscript reports results using a fixed balance factor selected through preliminary tuning. In the revision, we will include a comprehensive ablation study over multiple values of the balance factor (e.g., $\lambda \in \{0.1, 0.5, 1.0, 2.0, 5.0\}$) and analyze its impact on reconstruction quality, generation fidelity, and training stability. This will provide readers with better guidance on hyperparameter selection for different datasets and application scenarios.

---

### Author Response · Authors · 2025-12-02
**General Response To All Reviewers**

We thank the reviewers for their detailed and constructive feedback. In particular, we appreciate the insight regarding the discrepancy between our textual description of the forward process and the implemented algorithm.
Clarification on Architecture:
Based on the feedback, we recognize that our initial manuscript was ambiguous regarding the role of the quantum process. To resolve the mathematical inconsistencies pointed out regarding non-Gaussian distributions, we are revising the manuscript to explicitly frame our method as Quantum-Regularized Latent Diffusion rather than pure Quantum Diffusion.

Forward Phase 1 (Quantum Regularization): The PQC acts as a noisy feature map within the encoder. It injects quantum noise to contract the Lipschitz constant of the latent embedding and shrink the Quantum Fisher Information Matrix (QFIM), creating a robust latent manifold.
Forward Phase 2 (Gaussian Diffusion): The temporal diffusion process applied to this regularized latent is standard classical Gaussian diffusion.
Reverse Phase: Because the temporal corruption is Gaussian, the standard Score Matching objective is mathematically valid.

This revision aligns the text with Algorithm 1 and addresses theoretical concerns regarding the invertibility of quantum channels. We believe this clarification strengthens both the theoretical foundation and practical interpretation of our contribution.

---

### Note · Program_Chairs · 2026-01-17
**Submission Desk Rejected by Program Chairs**

The following references in this submission do not refer to real documents and/or have major errors in bibliographic information:

 1. David A McAllester. Some PAC-Bayesian complexity results for logistic regression. In Advances in Neural Information Processing Systems, volume 12, pp. 561-568, 1999.

2. Maria Schuld and Nathan Killoran. Effect of quantum noise on the generalization of quantum models. In QML Workshop, 2021.